# Using multiple sampling strategies to estimate SARS-CoV-2 epidemiological parameters from genomic sequencing data

Rhys P. D. Inward [1] ✉, Kris V. Parag [2,3,5] ✉ & Nuno R. Faria [1,2,4,5] ✉

The choice of viral sequences used in genetic and epidemiological analysis is important as it can induce biases that detract from the value of these rich datasets. This raises questions about how a set of sequences should be chosen for analysis. We provide insights on these largely understudied problems using SARS-CoV-2 genomic sequences from Hong Kong, China, and the Amazonas State, Brazil. We consider multiple sampling schemes which were used to estimate $R_t$ and $r_t$ as well as related $R_O$ and date of origin parameters. We find that both $R_t$ and $r_t$ are sensitive to changes in sampling whilst $R_O$ and the date of origin are relatively robust. Moreover, we find that analysis using unsampled datasets result in the most biased $R_t$ and $r_t$ estimates for both our Hong Kong and Amazonas case studies. We highlight that sampling strategy choices may be an influential yet neglected component of sequencing analysis pipelines.

Severe acute respiratory syndrome coronavirus 2 (SARS-CoV-2) is an enveloped single-stranded zoonotic RNA virus belonging to the *Betacoronavirus* genus and *Coronaviridae* family[1]. It was first identified in late 2019 in a live food market in Wuhan City, Hubei Province, China[2]. Within a month, SARS-CoV-2 had disseminated globally through sustained human-to-human transmission. It was declared a public health emergency of international concern on the 30 January 2020 by the World Health Organisation[3]. Those infected with SARS-CoV-2 have phenotypically diverse symptoms ranging from mild fever to multiple organ dysfunction syndromes and death[4].

Despite the implementation of non-pharmaceutical interventions (NPIs) and rollout of vaccination programmes in many countries to control their epidemics, as of the 16 July 2022, over 557 million SARS-CoV-2 cases and 6.3 million deaths have been reported worldwide[5]. These NPIs can vary within and between countries and include restrictions on international and local travel, school closures, social distancing measures, and the isolation of infected individuals and their contacts[6]. The key aim of NPIs is to reduce epidemic transmission, often measured by epidemiological parameters such as the time-varying effective reproduction number ($R_t$ at time $t$) and

growth rate ($r_t$), which both provide updating measures of the rate of spread of a pathogen (see Table 1 for detailed definitions)[7,8].

However, there is currently great difficulty in estimating and comparing epidemiological parameters derived from case and death data globally due to disparities in molecular diagnostic surveillance and notification systems between countries. Further, even if data are directly comparable, the choice of epidemiological parameter can implicitly shape insights into how NPIs influence transmission potential[9,10]. As such, there is a need to supplement traditional estimates with information derived from alternative data sources, such as genomic data[11], to gain improved and more robust insights into viral transmission dynamics[12,13].

Phylodynamic analysis of virus genome sequences have increasingly been used for studying emerging infectious diseases, as seen during the current SARS-CoV-2 pandemic[14–17], recent Ebola virus epidemics in Western Africa[18] and the Zika epidemic in Brazil and the Americas[19,20]. Transmissibility parameters such as the basic reproduction number ($R_O$), $R_t$ and $r_t$ can be directly inferred from genomic sequencing data or from epidemiological data, while other epidemiological parameters such as the time of the most recent common ancestor (TMRCA) of a given viral variant or lineage can only

[1]Department of Zoology, University of Oxford, Oxford, UK. [2]MRC Centre of Global Infectious Disease Analysis, Jameel Institute for Disease and Emergency Analytics, Imperial College London, London, UK. [3]NIHR Health Protection Research Unit in Behavioural Science and Evaluation, University of Bristol, Bristol, UK. [4]Instituto de Medicina Tropical, Faculdade de Medicina da Universidade de Sao Paulo, Sao Paulo, Brazil. [5]These authors jointly supervised this work: Kris V. Parag, Nuno R. Faria. ✉e-mail: rhys.inward@zoo.ox.ac.uk; k.parag@imperial.ac.uk; n.faria@imperial.ac.uk

**Table 1 | Key parameters and definitions for SARS-CoV-2**

| Parameter | Definition |
|---|---|
| Basic reproduction number ($R_O$) | Average number of individuals infected by a single infected person in a fully susceptible population |
| Time-varying or effective reproduction number ($R_t$) | Average number of secondary infections generated per effective primary case at a certain time point and in the presence of susceptible depletion or interventions |
| Growth rate ($r_t$) | Rate of change of the logarithm of the number of new cases (i.e., the case incidence) per unit of time |
| Incubation period | Time between infection and symptom onset |
| Infectious period | Period in which an infectious host can transmit infectious agents to a susceptible individual |
| Generation interval | Time between infection events in an infector–infectee pair |
| Time of the most recent common ancestor or origin time | Date in which viral variant is thought to have emerged |
| Serial Interval | Time between symptom onsets in an infector–infectee pair |

be estimated from genomic data. This is of particular importance for variants of concern (VOC), genetic variants with evidence of increased transmissibility, more severe disease, and/or immune evasion. VOC are typically detected through virus genome sequencing and only limited inferences can be made using epidemiological data alone[21].

Currently, SARS-CoV-2 virus genomes from COVID-19 cases are being sequenced at an unprecedented pace providing a wealth of virus genomic datasets[22]. There are currently over 11.9 million genomic sequences available on GISAID, an open-source repository for influenza and SARS-CoV-2 genomic sequences[23]. These rich datasets can be used to provide an independent perspective on pathogen dynamics and can help validate or challenge parameters derived from epidemiological data. Specifically, the genomic data can potentially overcome some of the limitations and biases that can result from using epidemiological data alone. For example, genomic data are less susceptible to changes at the government level such as alterations to the definition of a confirmed case and changes to notification systems[24,25]. Inferences from virus genomic data improve our understanding of underlying epidemic spread and can facilitate better-informed infection control decisions[26]. However, these advantages are not straightforward to realise. The added value of genomic data depends on two related variables: sampling strategy and computational complexity.

The most popular approaches used to investigate changes in virus population dynamics include the Bayesian Skyline Plot[27] and Skygrid[28] models and the Birth-Death Skyline (BDSKY)[29]. These integrate Markov Chain Monte Carlo (MCMC) procedures and often converge slowly on large datasets[30]. As such, currently available SARS-CoV-2 datasets containing thousands of sequences become computationally impractical to analyse and sub-sampling is necessary. Although previous studies have examined how sampling choices might influence phylodynamic inferences[30–34], this remains a neglected area of study[35], particularly concerning SARS-CoV-2 for which sequencing efforts have been unprecedented[36]. To our knowledge, there are no published studies concerning SARS-CoV-2 which explore the effect that sampling strategies can have on the phylodynamic reconstruction of key transmission parameters. Incorrectly implementing a sampling scheme or ignoring its importance can mislead inferences and introduce biases[30,37].

This raises the important question that motivates our analysis: how should sequences be selected for phylodynamic analysis and which parameters are sensitive or robust to changes in different sampling schemes. Here we explore how diverse sampling strategies in genomic sequencing may affect the estimation of key epidemiological parameters. We estimate $R_O$, $R_t$, and $r_t$ from genomic sequencing data under different sampling strategies from a location with higher genomic coverage represented by Hong Kong, and a location with lower genomic coverage represented by the Amazonas state, Brazil. We then compare our estimates against those derived from reference case data. By benchmarking genomic inferences against those from case data we can better understand the impact that sampling strategies may have on phylodynamic inference, bolster confidence in estimates

of genomic-specific parameters such as the origin time (or TMRCA) and improve the interpretation of estimates from areas with heterogeneous genomic coverage.

## Results

### Sampling schemes

**Hong Kong.** Hong Kong reacted rapidly upon learning of the emergence of SARS-CoV-2 in Wuhan, Hubei province, China, by declaring a state of emergency on the 25 January 2020 and by mobilising intensive surveillance schemes in response to initial cases[38]. This appeared to be successful in controlling the first wave of cases. However, due to imported cases from Europe and North America, a second wave of SARS-CoV-2 infections emerged prompting stricter NPIs such as the closure of borders and restrictions on gatherings[38]. Following these measures, the incidence of SARS-CoV-2 rapidly decreased (Fig. 1). Hong Kong has a high sampling intensity with 11.6% of confirmed cases sequenced during our study period. Further, Hong Kong has high quality case data with a high testing rate through effective tracing of close contacts, testing of all asymptomatic arriving travellers and all patients with pneumonia[39].

The number of cases within Hong Kong for each week was used to inform the sampling schemes used within this study. This resulted in the unsampled scheme having $N = 117$ sequences, the proportional sampling scheme having $N = 54$ sequences, the uniform sampling scheme having $N = 79$, and the reciprocal-proportional sampling scheme having $N = 84$ sequences (Supplementary Fig. 3).

**Amazonas.** The Amazonas state of Brazil had its first laboratory confirmed case of SARS-CoV-2 in March 2020 in a traveller returning from Europe[40]. After a first large wave of SARS-CoV-2 infections within the state that peaked in early May 2020 (Fig. 2), the epidemic waned, cases dropped, remaining stable until mid-December 2020. The number of cases then started growing exponentially, ushering in a second epidemic wave. This second wave peaked in January 2021 (Fig. 2) and coincided with the emergence of a new SARS-CoV-2 VOC, designated P.1/Gamma[14].

To combat this second wave, the Government of the Amazonas state suspended all non-essential commercial activities on the 23 December 2020 (http://www.pge.am.gov.br/legislacao-covid-19/). However, in response to protests, these restrictions were reversed, and cases continued to climb. On the 12th of January, when local transmission of P.1/Gamma was confirmed in Manaus, capital of Amazonas state[41], NPIs were re-introduced (http://www.pge.am.gov.br/legislacao-covid-19/) which seemed to be successful in reducing the case incidence in the state. However, cases remained comparatively high (Fig. 2). Amazonas has a sampling intensity with 2.4% of suspected P.1/gamma cases sequenced during our study period.

The number of cases within the Amazonas state informed the sampling schemes used within this study. This resulted in the unsampled scheme having $N = 196$ sequences, the proportional

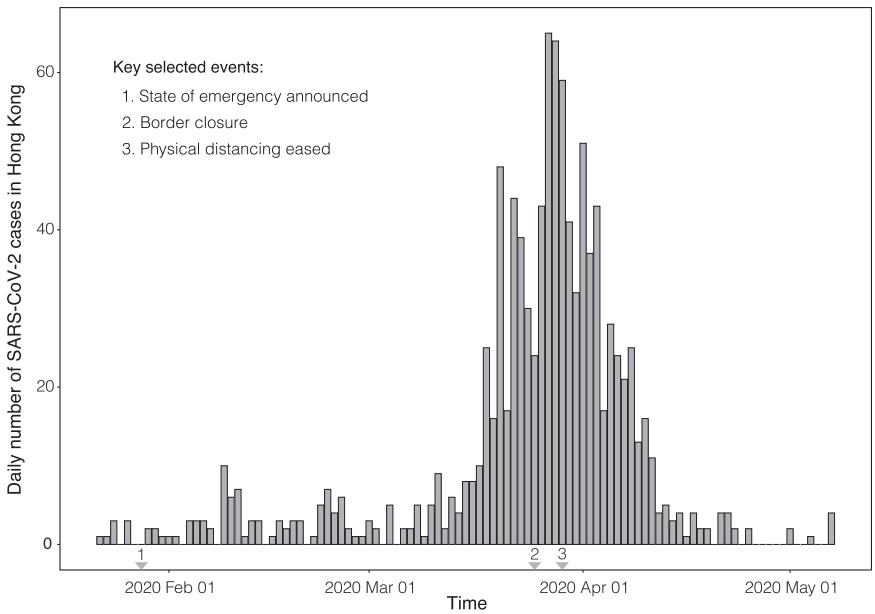

**Fig. 1 | Confirmed incident SARS-CoV-2 cases from Hong Kong until 7th of May 2020.** The arrows represent policy change-times[38].

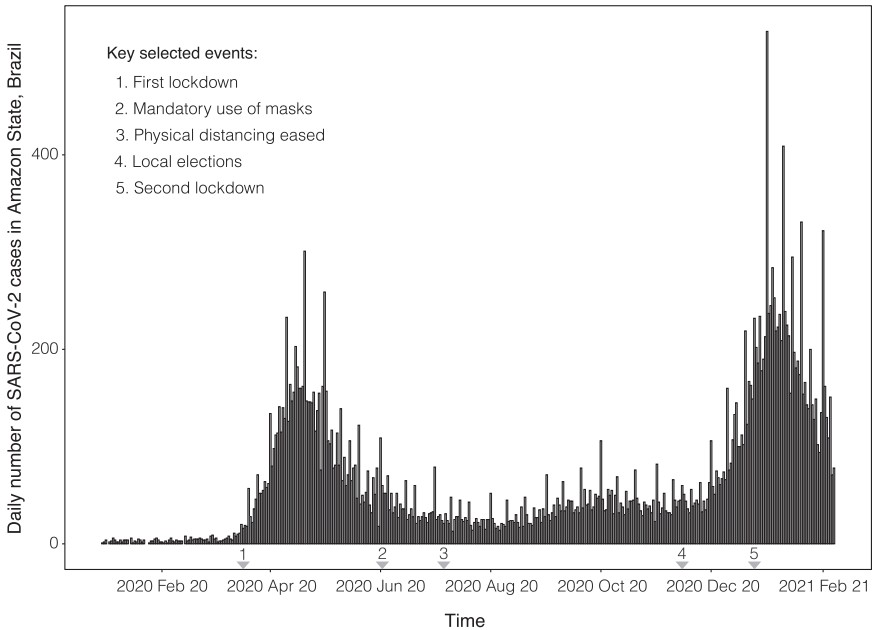

**Fig. 2 | Confirmed incident SARS-CoV-2 cases from Amazonas state, north Brazil until 7th of February 2021.** The arrows represent key policy change-times[52].

sampling scheme having $N = 168$ sequences, the uniform sampling scheme having $N = 150$, and the reciprocal-proportional sampling scheme having $N = 67$ sequences (Supplementary Fig. 4).

### Root-to-tip regression

The correlation ($R^2$) between genetic divergence and sampling dates for the Hong Kong datasets ranged between 0.36 and 0.52 and between 0.13 and 0.20 for the Amazonas datasets (Supplementary Fig. 2). This implies that the Hong Kong datasets have a stronger temporal signal. This is likely due to the Hong Kong datasets having a wider sampling interval (106 days) compared with the Amazonas datasets (69 days). A wider sampling interval can lead to a stronger temporal signal[42]. The gradient (rate) of the regression ranged from $1.16 \times 10^{-3}$ to $2.09 \times 10^{-3}$ substitutions per site per year (s/s/y) for the Hong Kong datasets and $4.41 \times 10^{-4}$ to $5.30 \times 10^{-4}$ s/s/y for the Amazonas datasets.

### Estimation of evolutionary parameters

The mean substitution rate (measured in units of number of s/s/y) and the TMRCA was estimated in BEAST, for both datasets, and the estimation from all sampling schemes was compared.

**Hong Kong.** For Hong Kong, the mean substitution rate per site per year ranged from $9.16 \times 10^{-4}$ to $2.09 \times 10^{-3}$ with sampling schemes all having overlapping Bayesian credible intervals (BCIs) (Supplementary Table 2 and Supplementary Fig. 5A). This indicates that the sampling scheme did not have a significant impact on the estimation of the clock rate. Moreover, the clock rate is comparable to estimations from the root-to-tip regression and to early estimations of the mean substitution rate per site per year of SARS-CoV-2[13].

Molecular clock dating of the Hong Kong dataset indicates that the estimated time of the most common recent ancestor was around

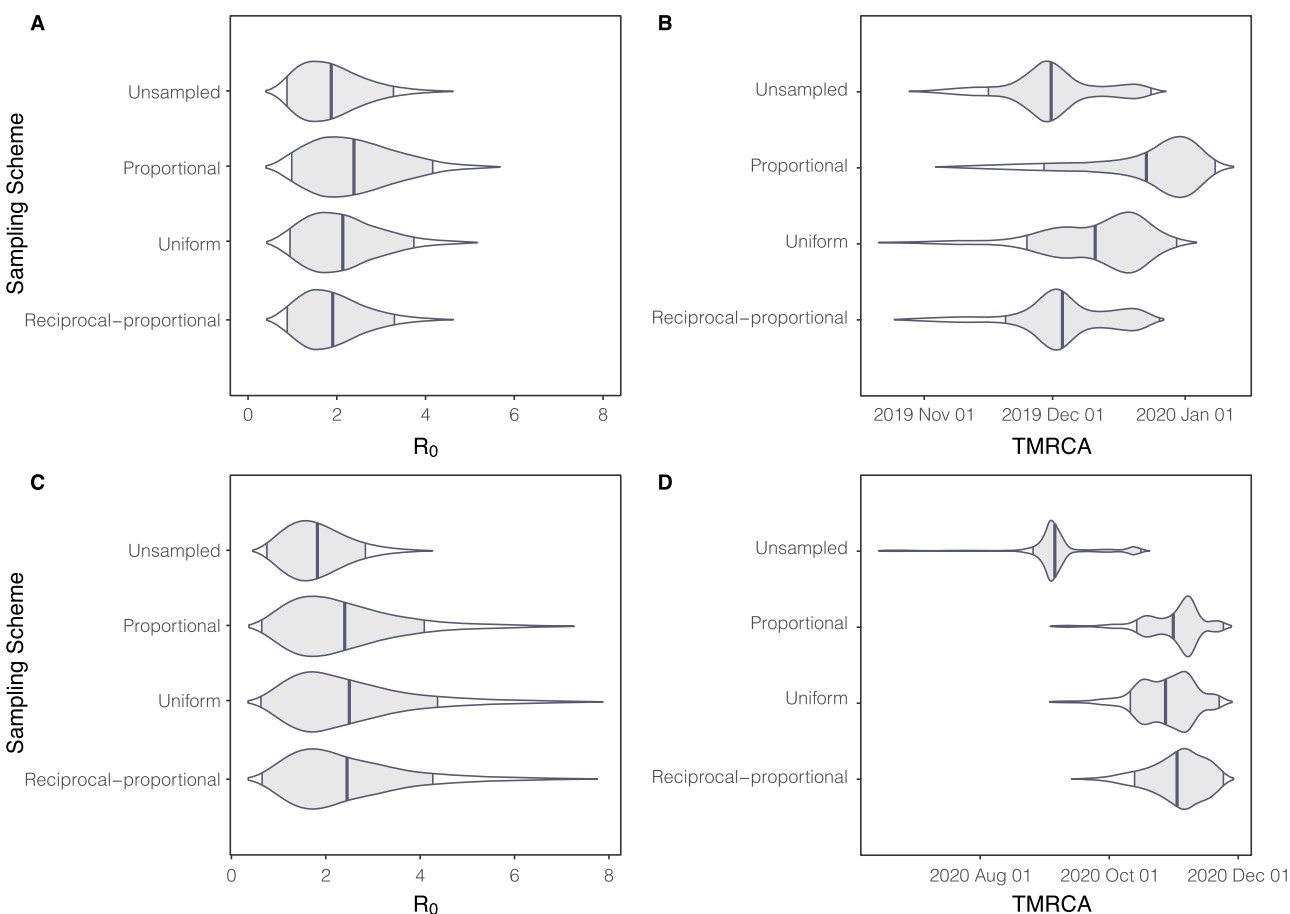

**Fig. 3 | $R_O$ estimated from BDSKY (using sequence data) and TMRCA for Hong Kong and Brazil. A, B** Represent Hong Kong and **C**, **D** represent the Amazonas State, Brazil. The central line represents the posterior mean and with intervals representing 95% highest posterior density interval.

December 2020 (Fig. 3B and Supplementary Table 2). This is a few weeks before the first confirmed case which was reported on the 18 January 2021. Once again, all sampling strategies have overlapped BCIs and with the range in means differing by around three weeks, a relatively short time scale, suggesting that the sampling scheme does not significantly impact the estimation of the TMRCA.

**Brazil.** For the Gamma VOC in the Amazonas state, the mean substitution rate ranged from $4.00 \times 10^{-4}$ to $5.56 \times 10^{-4}$ s/s/y with all sampling schemes having overlapped BCIs (Fig. 3D, Supplementary Table 2 and Supplementary Fig. 5B). This indicates that sampling strategy does not impact the estimation of the clock rate, supporting findings from the Hong Kong dataset. This also supports estimations from the root-to-tip analysis (Supplementary Fig. 2).

Molecular clock dating estimated a TMRCA mean around late October to early November (Fig. 3D and Supplementary Table 2). This is around five weeks before the date of the first P.1 case identified in Manaus used in our study. All sampling schemes have overlapping BCI consistent with the conclusion from the Hong Kong data that TMRCA is relatively robust to sampling.

**Estimation of basic reproduction number**
We found from using genomic data, Hong Kong had a posterior mean $R_O$ estimate of 2.07 (Fig. 3A) across all sampling strategies. Using a proportional sampling strategy gave the highest posterior mean $R_O$ estimate of 2.38 with the unsampled sampling strategy giving the lowest posterior mean $R_O$ estimate of 1.87. Overall, Brazil had a higher posterior mean $R_O$ estimate with a value of 2.24 (Fig. 3B) across all sampling strategies. The uniform sampling strategy yielded the

highest posterior mean $R_O$ estimate of 2.50 while the unsampled sampling strategy gave the lowest one of 1.82. Using case data, we found similarly found that Hong Kong had a lower $R_O$ of 2.17 (95% credible interval (CI) = 1.43–2.83) when compared with Amazonas which had a $R_O$ of 3.67 (95% CI = 2.83–4.48). All sampling schemes for both datasets were characterised by similar $R_O$ values (Fig. 3) indicating that the estimation of $R_O$ is robust to changes in sampling scheme.

**Time-varying reproduction number and growth rate**
We estimate $R_t$ and $r_t$ for local SARS-CoV-2 epidemics in Hong Kong and Amazonas, Brazil. Our main results showing these two parameters and JSD metrics are shown in Figs. 4–7.

**Hong Kong.** We applied the BDSKY model to estimate the $R_t$ for each dataset subsampled according to the different sampling strategies (Fig. 4). We compared these against the $R_t$ from incidence data, derived from *EpiFilter*. Based on the proportional sampling scheme, which had the lowest JSD (Fig. 4E), we initially infer a super-critical $R_t$ value, with a mean ~2, that appears to fall swiftly in response to the state of emergency and the rapid implementation of NPIs. A steady transmission rate subsequently persisted throughout the following weeks around the critical threshold ($R_t$ = 1). This period is followed by a sharp increase in $R_t$, peaking at a mean value of 2.6. This is likely due to imported cases from North America and Europe[38]. This led to a ban on international travel resulting in a sharp decline in $R_t$ (Fig. 1). However, this decline lasted around a week with the mean $R_t$ briefly increasing until more stringent NPIs such as the banning of major gatherings were implemented. Following this, the $R_t$ continued its

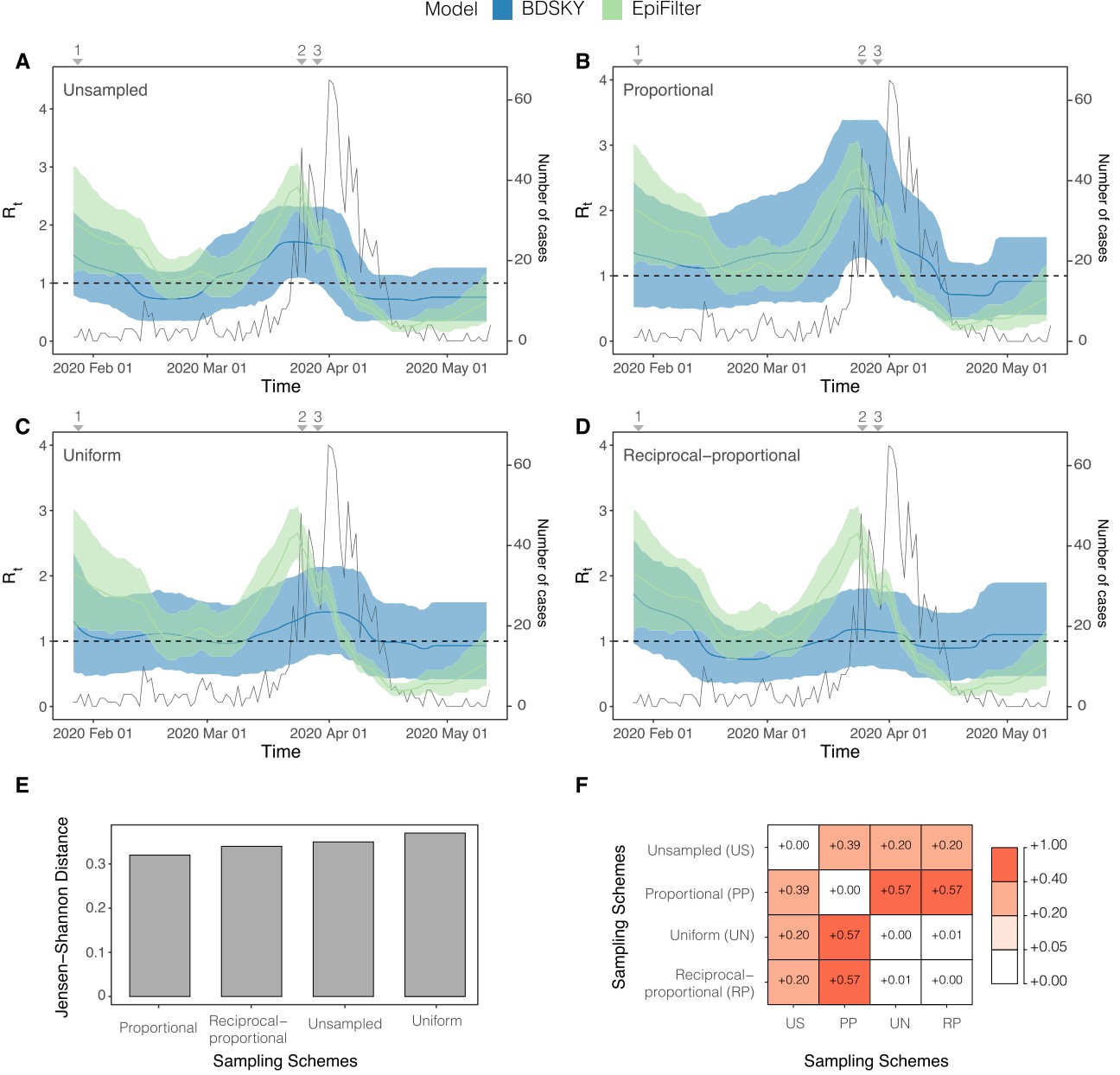

**Fig. 4 | $R_t$ estimated from both the Birth Death Skyline (BDSKY) and *EpiFilter* methods for Hong Kong.** Titles indicate the sampling scheme used in panels **A**–**D**. The light-shaded area represents the 95% highest posterior density interval. The solid line represents the mean $R_t$ estimate with *EpiFilter* in green and BDSKY in blue. The black line plots the number of cases. We refer to Fig. 1 for a brief description of key events 1–3. The Jensen Shannon Distance (JSD) is given in **E** and ranks the sampling strategies based on how similar the BDSKY estimates under those strategies are to those derived from *EpiFilter* (smaller values are better). **F** provides the pairwise JSD between the BDSKY estimates under different sampling strategies, showing often appreciable difference among strategies.

sharp decline falling below the critical threshold, with transmission becoming sub-critical (Fig. 4). The proportional sampling scheme showed the most divergence from all other sampling schemes whilst the uniform and reciprocal-proportional sampling schemes were almost identical (Fig. 4F).

These results were mirrored in the estimation of $r_t$ (Fig. 5), where estimates derived from the proportional sampling scheme showed the least divergence (Fig. 5E). There was an initial decline in the $r_t$, which steadied at a value of ~0, indicating that epidemic stabilisation had occurred. This stable period is followed by an increase in $r_t$ peaking at around a 0.050 per day (Fig. 5B). In response to NPIs, the $r_t$ starts to decrease, falling below 0, indicating a receding epidemic. The rate of this decline peaks at ~−0.075 per day (Fig. 5B). Unlike the estimation of $R_t$ (Fig. 4), the unsampled sampling scheme showed

the most divergence from all other sampling schemes (Fig. 5F). It also has a high divergence from estimates derived from *EpiFilter* when compared the proportional sampling scheme which was the most closely related to *EpiFilter* (Fig. 5E). Once again, the uniform and reciprocal-proportional schemes are the most closely related (Fig. 5E).

**Brazil.** The uniform, reciprocal-proportional, and proportional sampling schemes all showed a similarly low JSD (Fig. 6E). Based on these sampling schemes, we initially infer super-critical transmission ($R_t > 1$) with a mean value of 3 (Fig. 6). From this point, the $R_t$ declines, although it remains above the critical threshold ($R_t = 1$) for much of the study period. Sub-critical transmission ($R_t < 1$) was only reached after the re-imposition of NPIs. This implies that initial restrictions, such as

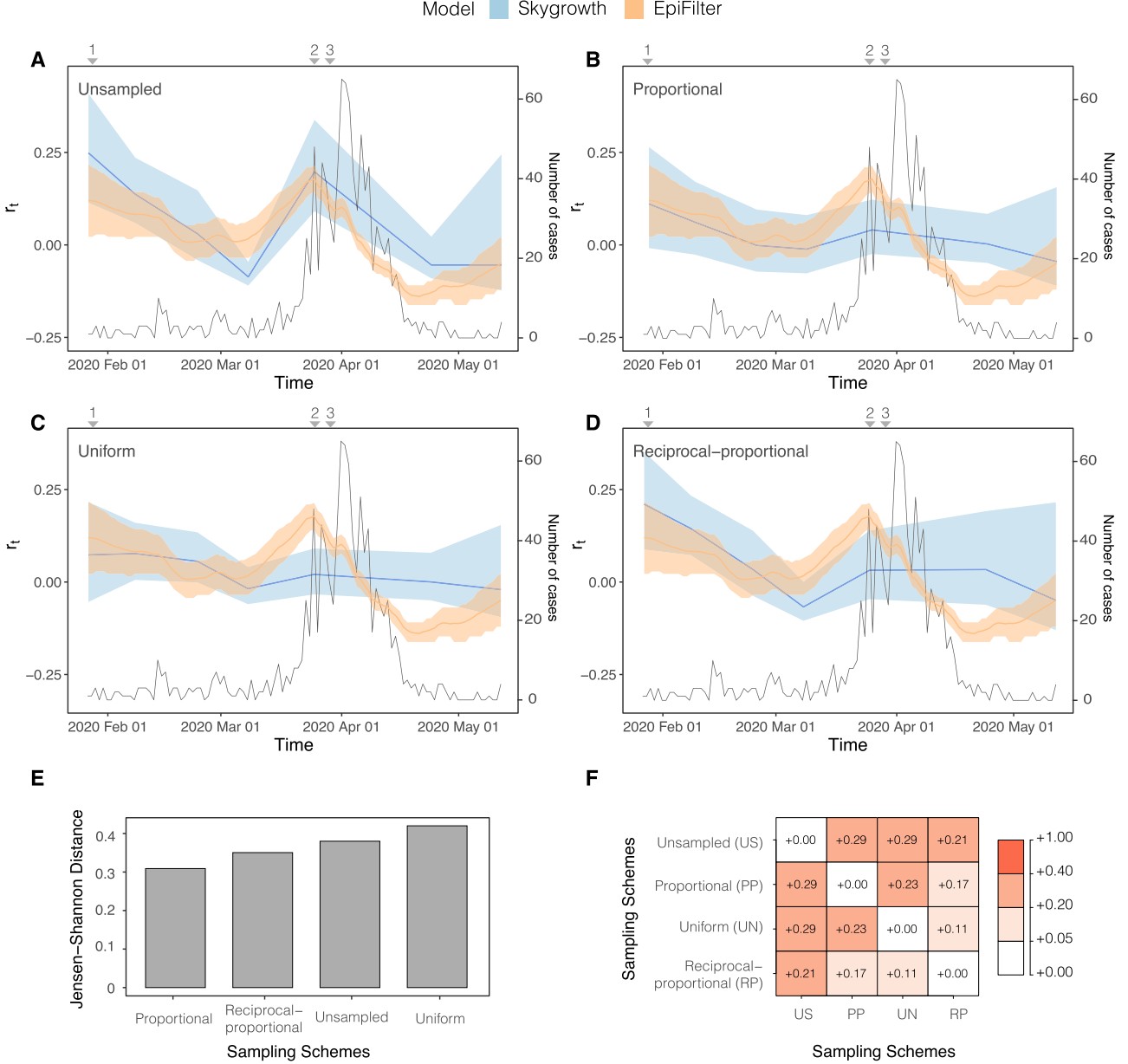

**Fig. 5 | $r_t$ estimated from both the *Skygrowth* and *EpiFilter* methods for Hong Kong.** Titles indicate the sampling scheme used in panels **A**–**D**. The light-shaded area represents the 95% highest posterior density interval. The solid line represents the mean $r_t$ estimate with *EpiFilter* in orange and *Skygrowth* in blue. The black line refers to the number of cases. We refer to Fig. 1 for a brief description of key events 1–3. The Jensen Shannon Distance (JSD) is given in **E** and ranks the sampling strategies based on how similar the *Skygrowth* estimates under those strategies are to those derived from *EpiFilter* (smaller values are better). **F** provides the pairwise JSD between the *Skygrowth* estimates under different sampling strategies, showing often appreciable difference among strategies.

the suspension of commercial activities, were likely insufficient for suppressing spread. Only after more stringent restrictions were imposed did $R_t$ become sub-critical. However, there is no evidence of a sharp decrease in $R_t$ once restrictions were re-imposed, which may suggest limited effectiveness. The unsampled sampling scheme again showed the most divergence from all other sampling schemes (Fig. 6F) and the highest divergence from the case data estimate (Fig. 6E) with the uniform and proportional sampling schemes showing the most similarity. As such, applying no sampling strategy/opportunistic sampling leads to, from the perspective of comparing to *EpiFilter*, the most biased estimates.

Based on the proportional sampling scheme, which had the lowest JSD (Fig. 7E) we infer a steady decline in $r_t$ which matches the pattern seen with the $R_t$ value (Fig. 7). The initial $r_t$ implied a 0.250 per day. Subsequently, the $r_t$ falls over the study period. $r_t$ falls below

0 after the re-imposition of NPIs declining at −0.030 per day by the end of the study period. There is no evidence of any noticeable declines in $r_t$ when interventions were introduced indicating that they might not have significantly impacted the growth rate of P.1/ gamma. The unsampled sampling scheme was again most divergent from other sampling schemes as well as from estimates derived from *EpiFilter* with the uniform and reciprocal-proportional being most similar.

## Discussion
In this study, we applied phylodynamic methods to available SARS-CoV-2 sequences from Hong Kong and the Amazonas state of Brazil to infer their key epidemiological parameters and to compare the impact that various sampling strategies have on the phylodynamic reconstruction of these parameters.

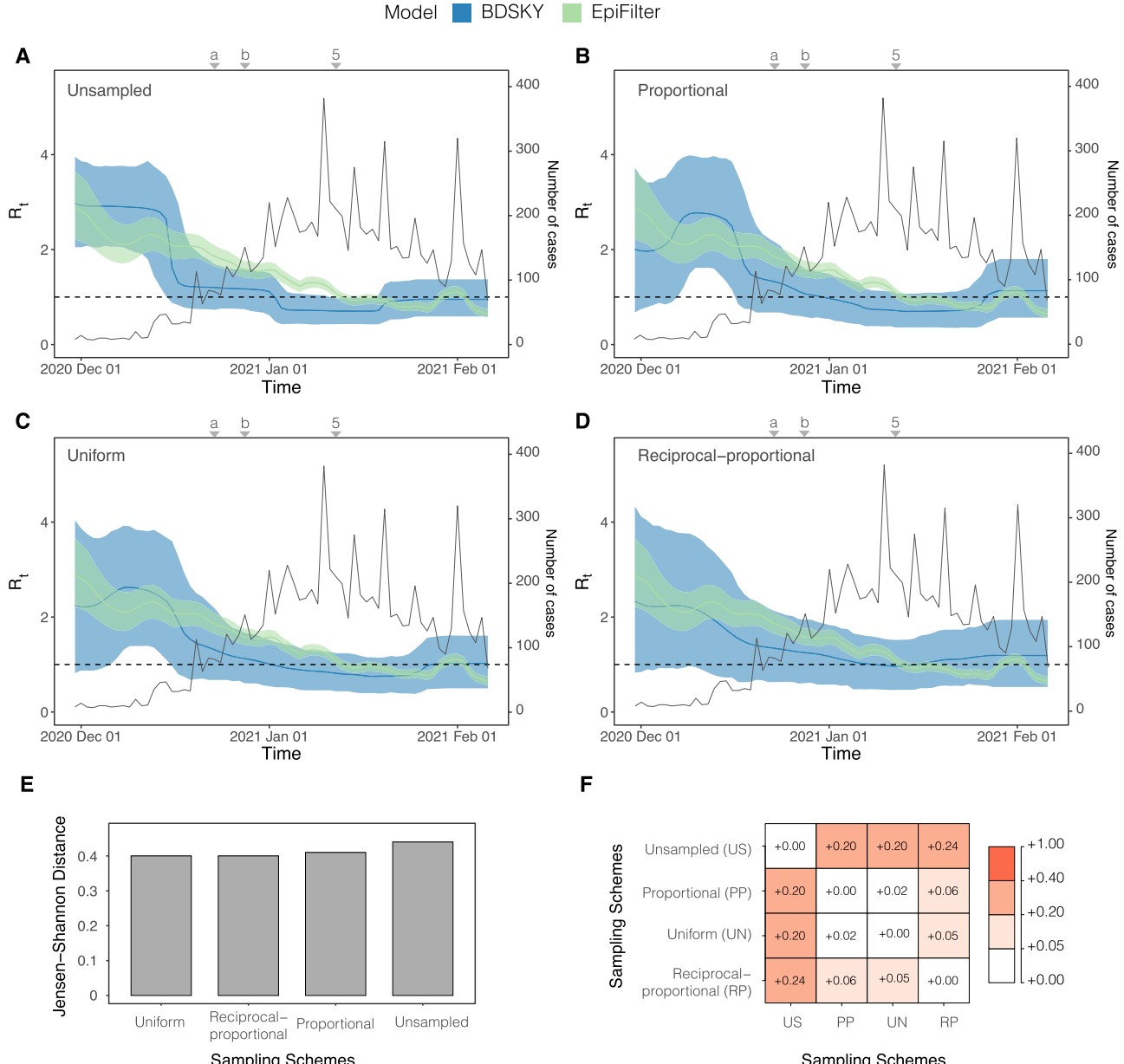

**Fig. 6 | $R_t$ estimated from both the Birth Death Skyline (BDSKY) and *EpiFilter* methods for Amazonas, Brazil.** Titles indicate the sampling scheme used in panels **A**–**D**. The light-shaded area represents the 95% highest posterior density interval. The solid line represents the mean $R_t$ estimate with *EpiFilter* in green and BDSKY in blue. We refer to Fig. 2 for a brief description of key events, including 5 which corresponds to the second lockdown. Event "a" corresponds to the suspension of commercial activities in Manaus; event "b" corresponds to the resumption of commercial activities in Manaus[52]. The Jensen Shannon Distance (JSD) is given in panel **E** and ranks the sampling strategies based on how similar the BDSKY estimates under those strategies are to those derived from *EpiFilter* (smaller values are better). Panel **F** provides the pairwise JSD between the BDSKY estimates under different sampling strategies, showing often appreciable difference among strategies.

We estimated the basic reproductive number of SARS-CoV-2 in Hong Kong to be 2.17 (95% CI = 1.43–2.83). This supports previous estimates of the initial $R_O$ in Hong Kong[38,43] which estimated $R_O$ to be 2.23 (95% CI = 1.47–3.42). For the Amazonas state in Brazil, we estimated the $R_O$ to be 3.67 (95% CI = 2.83 – 4.48). Although the population of Amazonas State may not be fully susceptible to P.1/Gamma[14,44], this should not affect the comparison among sampling schemes. We found that $R_O$ is robust to changes in sampling schemes (Fig. 3A, C).

For the Hong Kong dataset, the proportional sampling scheme was superior to all other sampling schemes in estimating $R_t$. It successfully predicted the initial super-critical $R_t$, its decline in response to rapid NPIs, and subsequent increase and decline during the second wave of infections (Fig. 4B). This was in comparison to the uniform sampling scheme, which provided the worst (largest) JSD (Fig. 4D) and an $R_t$ estimate that was largely insensitive to NPIs. The proportional sampling scheme, alongside the uniform sampling scheme, best estimated $r_t$ (Fig. 5B, C). In contrast, for the Amazonas dataset, the uniform sampling scheme best estimated the $R_t$ and $r_t$ (Fig. 6C) whilst the proportional sampling scheme best captured $r_t$ (Fig. 7C). It captured both its initial super-critical $R_t$ and high $r_t$ alongside their subsequent decline.

We found that estimates from all sampling schemes were distinct from those obtained using the unsampled data and that on some instances the sampling schemes were also appreciably different from one another (see panel F in Figs. 4–7) with the uniform and reciprocal-proportional sampling strategies being most similar. This highlights

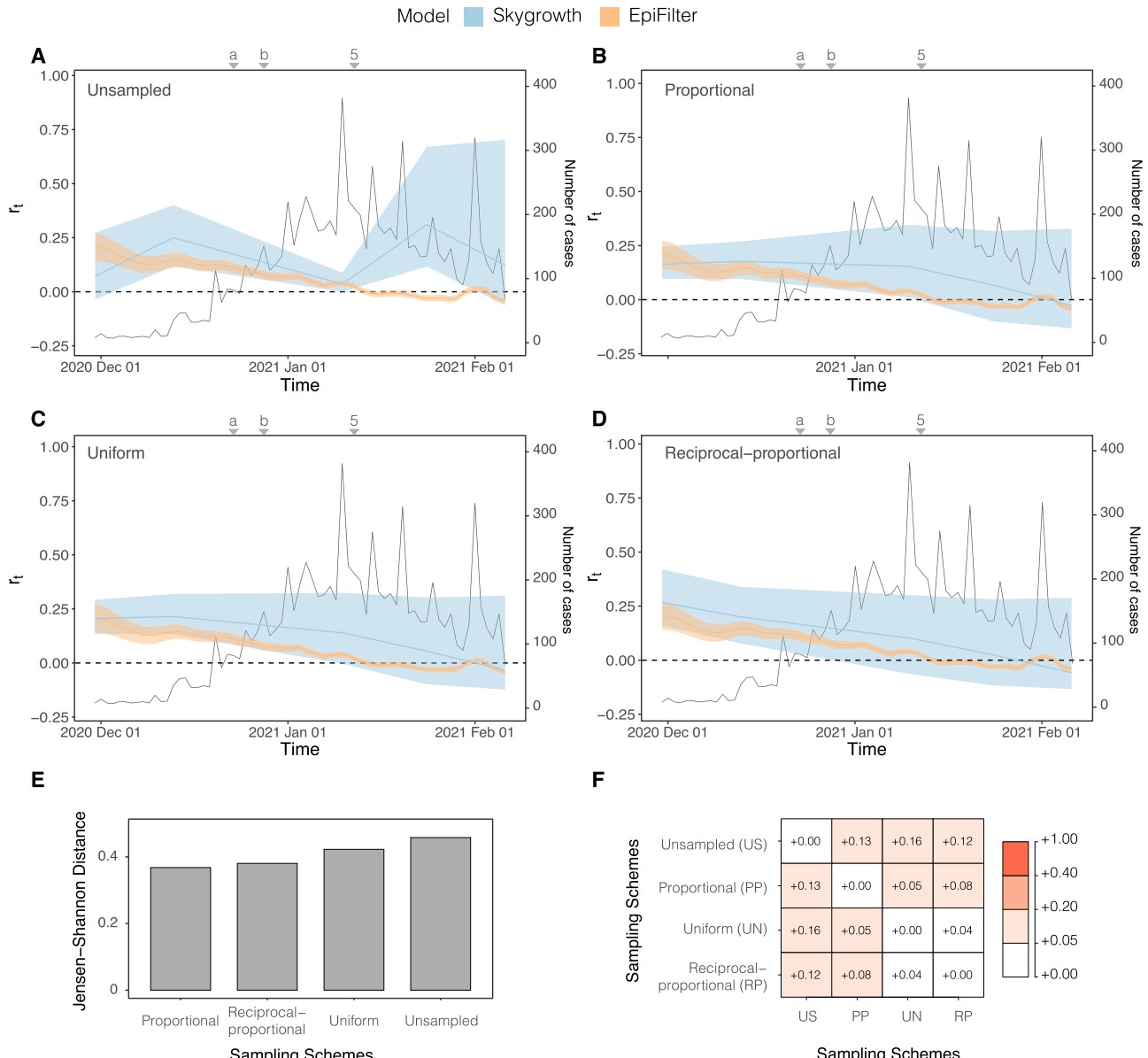

**Fig. 7 | $r_t$ estimated from both the *Skygrowth* and *EpiFilter* methods for Amazonas, Brazil.** Titles indicate the sampling scheme used in panels **A**–**D**. The light-shaded area represents the 95% highest posterior density interval. The solid line represents the mean $r_t$ estimate with *EpiFilter* in orange and *Skygrowth* in blue. We refer to Fig. 2 for a brief description of key events, including 5 which corresponds to the second lockdown. Event "a" corresponds to the suspension of commercial activities in Manaus; event "b" corresponds to the resumption of commercial activities in Manaus[52]. The Jensen Shannon Distance (JSD) is given in panel **E** and ranks the sampling strategies based on how similar the *Skygrowth* estimates under those strategies are to those derived from *EpiFilter* (smaller values are better). Panel **F** provides the pairwise JSD between the *Skygrowth* estimates under different sampling strategies, showing often appreciable difference among strategies.

how different sampling schemes can produce significantly differing estimates of epidemiological parameters and underscores the need for considering sampling and its potential impact on estimations.

Our $R_t$ estimates are consistent with previous estimates of Gamma VOC's transmissibility in Amazonas state[14]. This contrasted with the unsampled data in which the $r_t$ increased at the end of the period (Fig. 7A). This highlights that unlike $R_O$, both $R_t$ and $r_t$ are sensitive to changes in sampling and that even related epidemiological parameters like $R_t$ and $r_t$ may require different sampling strategies to optimise inferences.

Molecular clock dating of the Hong Kong and Amazonas dataset has revealed that the date of origin is relatively robust to changes in sampling schemes. For Hong Kong, SARS-CoV-2 likely emerged in mid-December 2019 ~5 weeks before the first reported case on the 22 January 2020[38]. The Amazonas dataset revealed that the date of the

common ancestor of the P.1 lineage emerged around late October 2020 to early November, ~5 weeks before the first reported case on the 6th of December[14], with all BCI's overlapping for each sampling strategy. Like the molecular clock dating, we found that the molecular clock rate was robust to changes in sampling strategies in both datasets with all sampling strategies having overlapped BCI's (Supplementary Table 2 and Supplementary Fig. 5). For the Hong Kong dataset, its clock rate is comparable to early estimations of the mean substitution rate per site per year of SARS-CoV-2[13]. However, the clock rate estimated for the Brazilian dataset is lower than the initial $8.00 \times 10^{-4}$ s/s/y which is used in investigating SARS-CoV-2[45] and that has been used in previous analyses of Gamma VOC[46]. This initial estimation of evolutionary rate was estimated from genomic data taken over a short time span at the beginning of the pandemic introducing a time dependency bias. By using a more appropriate

clock rate it can improve tree height and rooting resulting in more robust parameter estimations[47].

Treating sampling times as uninformative has been shown to be inferior to including them as dependent on effective population size and other parameters by several previous studies[30,31,34,48]. Although these studies did not consider the estimation of epidemiological parameters, they highlight the potential of systematic biases being introduced into the phylodynamic reconstruction by not using a sampling scheme or by assuming an incorrect model for how sampling schemes introduce information. This was supported by our results as phylodynamic inferences with no sampling strategy applied had the poorest overall performance for both Hong Kong and the Amazonas state. This implies that sampling design choices can significantly impact phylodynamic reconstruction, and that exploration of sampling strategies is needed to obtain the most reliable estimates of key epidemiological parameters.

Although our results provide rigorous insight into the dynamics of SARS-CoV-2 and the impact of sampling strategies in the Amazonas state and Hong Kong, there are limitations. The *Skygrowth* and BDSKY models do not explicitly consider imports into their respective regions. This is particularly relevant for Hong Kong as most initial sequences from the region were sequenced from importation events[49] which can introduce error into parameter estimation[50]. However, as the epidemic expanded, more infections were attributable to autochthonous transmission[49], and the risk of error introduced by importation events decreased. Moreover, while sampling strategies can account for temporal variations in genomic sampling fractions there is currently no way to account for non-random sampling approaches in either the BDSKY or *Skygrowth* models[51]. It is unclear how network-based sampling may affect parameter estimates obtained through these models[44] presenting a key challenge in molecular and genetic epidemiology. Spatial heterogeneities were also not explored within this work. This represents the next key step in understanding the impact of sampling as spatial sampling schemes would allow the reconstruction of the dispersal dynamics and estimation of epidemic overdispersion (*k*), a key epidemiological parameter.

Finally, we compared our phylodynamic estimates against epidemiological inferences derived from incident case data from Hong Kong and Amazonas state, two settings with very different diagnostic capacity. While Hong Kong has high quality case data with a high testing rate[39], there is a large underreporting of SARS-CoV-2 cases in the Amazonas state[52,53]. Future epidemiological modelling work is needed to compare parameter estimates obtained from case data, death data, and excess death data across different settings. This will improve the benchmarks we use to compare sequence-based estimates against.

This work has highlighted the impact and importance that applying temporal sampling strategies can have on phylodynamic reconstruction. Although more genomic datasets from a variety of countries and regions with different sampling intensities and proportions are needed to create a more generalisable sampling framework and to dissect any potential cofounders, this study has demonstrated that genomic datasets that commonly feature opportunistic sampling (i.e., there is no deliberate strategy design) can introduce significant uncertainty and biases in the estimation of epidemiological parameters. This finding signifies the need for more targeted attempts at performing genomic surveillance and epidemic analyses particularly in resource-poor settings with limited genomic capability.

## Methods
### Empirical estimation of the reproduction number, time-varying effective reproduction number, and growth rate
**Epidemiological datasets.** Two sources of data from the Amazonas state, Brazil, and one source of data from Hong Kong were used to calculate empirical epidemiological parameters. For the Amazonas

**Table 2 | Parameter estimates used within the deterministic SEIR model**

| Parameter | Description | Value (source) |
|---|---|---|
| $R_O = \beta\alpha$ | Basic Reproduction Number | Estimated |
| $N$ | Population of Hong Kong | 7,481,800 persons[80] |
| | Population of Amazonas state | 4,207,714 persons[81] |
| $\beta$ | Effective Contact Rate | Estimated |
| $\alpha$ | Infectious Period | 0.07 per day[82] |
| $\lambda$ | Force of Infection | Estimated |
| $\gamma$ | Progression from E to I | 5.26 per day[83] |
| $\sigma$ | Progression from I to R | 14.3 per day[82] |
| S | Estimated number of Susceptibles | Estimated |
| E | Estimated number of Exposed | Estimated |
| I | Number of Infected | Weekly case counts |
| R | Estimated number of Recovered | Estimated |

state, case data from the SIVEP-Gripe (Sistema de Informação de Vigilância Epidemiológica da Gripe) SARI (severe acute respiratory infections) database from the 30 November 2020 up to 7 February 2021 were used. Here we were interested in cases caused by the Gamma VOC first detected in Manaus[14]. The number of Gamma cases was calculated by using the proportion of Gamma viral sequences uploaded to GISAID within each week (Supplementary Fig. 1). For Hong Kong, all case data were extracted from the Centre of Health Protection, Department of Health, the Government of the Hong Kong Special Administrative region up to the 7 May 2020. Owing to lags in the development of detectable viral loads, symptom onset and subsequent testing[54]; the date on which each case was recorded was left shifted by 5 days within our models[55] to account for these delays in both datasets.

**Basic reproduction number.** The $R_O$ parameter was estimated using a time series of confirmed SARS-CoV-2 cases from both Hong Kong and the Amazonas state. To avoid the impact of NPIs, only data up to the banning of mass gathering in Hong Kong (27 March 2020) and until the imposition of strict restrictions in the Amazonas state (12 January 2021) were used. We estimated $R_O$ from weekly counts of confirmed cases using maximum likelihood methods. The weekly case counts were assumed to be Poisson distributed and were fitted to a closed Susceptible-Exposed-Infectious-Recovered (SEIR) model (Eq. (1)) by maximising the likelihood of observing the data given the model parameters (Table 2). Subsequently, the log-likelihood was used to calculate the $R_O$ by fitting $\beta$, the effective contact rate.

$$\lambda = \frac{\beta I}{N} \frac{dS}{dt} = -\lambda S \frac{dE}{dt} = \lambda S - \gamma E \frac{dI}{dt} = \gamma E - \sigma I \frac{dR}{dt} = \sigma I \qquad (1)$$

To generate ~95% confidence intervals (CIs) for $R_O$, non-parametric bootstrapping was used with 1000 iterations.

**Time-varying effective reproduction number.** To estimate $R_t$ from case time series data the *EpiFilter* method[56] was used. *EpiFilter* describes transmission using a renewal model; a general and popular framework that can be applied to infer the dynamics of numerous infectious diseases from case incidence[57]. This model describes how the number of new cases (incidence) at time *t* depends on $R_t$ at that specified time point and the past incidence, which is summarised by the cumulative number of cases up to each time point weighted by the generation time distribution. *EpiFilter* integrates both Bayesian forward and backward recursive smoothing. This improves $R_t$ estimates by leveraging the benefits of two of the most popular $R_t$

estimation approaches: *EpiEstim*[58] and the Wallinga-Teunis method[59]. *EpiFilter* minimises the mean squared error in estimation and reduces dependence on prior assumptions, making it a suitable candidate for deriving reference estimates. We use these to benchmark estimates independently obtained from genomic data. We assume the generation time distribution is well approximated by the serial interval (SI) distribution[58], which describes the times between symptom onsets between an infector–infectee pair. We describe the specific SI distributions that we used in the next subsection.

**Growth rate.** After $R_t$ has been inferred, the Wallinga-Lipsitch equation for a gamma distributed generation time distribution (Eq. (2)) was used to estimate the exponential epidemic $r_t$[60]. The SI for Hong Kong was derived from a systematic review and meta-analysis[61] and a study exploring SI in Brazil was used for the Amazonas datasets[62]. The SI was assumed to be gamma distributed. The gamma distribution is represented by gamma ($\varepsilon$, $\gamma$) with $\varepsilon$ and $\gamma$ being the shape and scale parameters respectively.

$$r_t = \varepsilon \left( R_t^{\left(\frac{1}{\gamma}\right)} - 1 \right) \qquad (2)$$

### SARS-CoV-2 Brazilian Gamma VOC and Hong Kong datasets
All high-quality (<1% N, or non-identified nucleotide), complete (>29 kb) SARS-CoV-2 genomes were downloaded from GISAID[23] for Hong Kong (up to 7 May 2020) and the Amazonas state, Brazil (from 30 November 2020 up to 7 February 2021). Using the Accession ID of each sequence, all sequences were screened and only sequences previously analysed and published in PubMed, MedRxiv, BioRxiv, virological, or Preprint repositories were selected for subsequent analysis. For both datasets, sequence alignment was conducted using MAFFTV.7[63]. The first 130 base pairs (bp) and last 50 bps of the aligned sequences were trimmed to remove potential sequencing artefacts in line with the Nextstrain protocol[64]. Both datasets were then processed using the Nextclade pipeline for quality control (https://clades.nextstrain.org/). Briefly, the Nextclade pipeline examines the completeness, divergence, and ambiguity of bases in each genetic sequence. Only sequences deemed 'good' by the Nextclade pipeline were selected. Subsequently, all sequences were screened for identity and in the case of identical sequences, for those with the same location, collection date, only one such isolate was used. Moreover, PANGO lineage classification was conducted using the Pangolin[22] (v2.3.3) software tool (http://pangolin.cog-uk.io) on sequences from the Amazonas state and only those with the designated P.1/Gamma lineage were selected (Supplementary Figure 1).

### Maximum Likelihood tree reconstruction
Maximum likelihood phylogenetic trees were reconstructed using IQTREE2[65] for both datasets. A TIM2 model of nucleotide substitution with empirical base frequencies and a proportion of invariant sites was used as selected for by the ModelFinder application[66] for the Hong Kong dataset. For the Brazilian dataset, a TN model of nucleotide substitution[67] with empirical base frequencies was selected for. To assess branch support, the approximate likelihood-ratio test based on the Shimodaira–Hasegawa-like procedure with 1,000 replicates[68], was used.

### Root-to-tip regression
To explore the temporal structure of both the Brazilian and Hong Kong dataset, TempEst v.1.5.3[69] was used to regress the root-to-tip genetic distances against sampling dates (yyyy-mm-dd). The 'best-fitting' root for the phylogeny was found by maximising the $R^2$ value of the root-to-tip regression (Supplementary Fig. 2). Several sequences showed incongruent genetic diversity and were discarded from subsequent analyses. This resulted in a final dataset of $N = 117$ Hong Kong

sequences and $N = 196$ Brazilian sequences. The gradient of the slopes (clock rates) provided by TempEst were used to inform the clock prior in the phylodynamic analysis.

### Subsampling for analysis
Four retrospective sampling schemes were used to select a subsample of Amazonas and Hong Kong sequences. Each sampling period was broken up into weeks with each week being used as an interval according to a temporal sampling scheme (without replacement). This temporal sampling scheme was based on the number of reported cases of SARS-CoV-2.

The temporal sampling schemes that we explored were:
- No sampling strategy applied: all sequences were included without a sampling strategy applied (equivalent to opportunistic sampling).
- Proportional sampling: weeks are chosen with a probability proportional to the value of the number of incident cases in each epi-week.
- Uniform sampling: all weeks have equal probability.
- Reciprocal-proportional sampling: weeks are chosen with a probability proportional to the reciprocal of the incident number of cases in each epi-week.

These sampling schemes were inspired by those recommended by the WHO for practical use in different settings and scenarios[70]. Proportional sampling is equivalent to representative sampling, uniform sampling is equivalent to fixed sampling whilst the unsampled data includes all sampled sequences. Reciprocal-proportional sampling is not commonly applied in practice and was used as a control within this study.

### Bayesian evolutionary analysis
Date molecular clock phylogenies were inferred for all sampling strategies applied to the Amazonas and Hong Kong dataset using BEAST v1.10.4[71] with BEAGLE library v3.1.0[72] for accelerated likelihood evaluation. For both the Amazonas and Hong Kong datasets, a HKY substitution model with gamma-distributed rate variation among sites and four rate categories was used to account for among-site rate variation[73]. A strict clock molecular clock model was chosen. Both the Amazonas and Hong Kong dataset were analysed under a flexible non-parametric skygrid tree prior[74]. Four independent MCMC chains were run for both datasets. For the Amazonas dataset, each MCMC chain consisted of 250,000,000 steps with sampling every 50,000 steps. Meanwhile, for the Hong Kong dataset, each MCMC chain consisted of 200,000,000 steps with sampling every 40,000 steps. For both datasets, the four independent MCMC runs were combined using LogCombiner v1.10.4[71]. Subsequently, 10% of all trees were discarded as burn in, and the effective sample size of parameter estimates were evaluated using TRACER v1.7.2[75]. An effective sample size of over 200 was obtained for all parameters. Maximum clade credibility (MCC) trees were summarised using Tree Annotator[71].

### Phylodynamic reconstruction
**Estimation of the basic and time-varying effective reproduction numbers.** The Bayesian birth-death skyline (BDSKY) model[29] implemented within BEAST 2 v2.6.5[76] was applied to estimate the time-varying transmissibility parameter $R_t$ (Table 3). An HKY substitution model with a gamma-distributed rate variation among sites and four rate categories[73] was used alongside a strict molecular clock model. The selected number of intervals for both datasets was 5, representing $R_t$ changing every 2.5 weeks for the Hong Kong datasets and every 2 weeks for the Brazilian datasets, with equidistant intervals per step. An exponential distribution was used with a mean of 36.5 per year for the rate of becoming infectious, assuming a mean duration of infection

**Table 3 | Values and priors for the parameters of the BDSKY model. s/s/y=substitutions per site per year**

| Parameter | Dataset | Value or prior | Rationale/Assumption |
|---|---|---|---|
| Clock rate | Brazil | $4.0 \times 10^{-4}$ s/s/y | Informed by root-to-tip regression |
| | Hong Kong | $1.0 \times 10^{-4}$ s/s/y | |
| Death rate | Brazil and Hong Kong | 36.5 $y^{-1}$ | The period between infection and becoming uninfectious assumed an exponential distribution with a mean of 10 days[15] |
| Reproduction number | Brazil and Hong Kong | Lognormal (0.8, 0.5) | Median 2.2, 95% IQR 0.8–5.9 |
| Time of origin | Brazil | Lognormal (−1.50, 0.4) y before present | Median 4 December 2020, 95% IQR 25 September 2020 to 12 January 2021 |
| | Hong Kong | Lognormal (−1.75, 0.4) y before present | Median 18 January 2020, 95% IQR 17 November 2019 to 15 February 2020 |
| Sampling proportion | Brazil | Uniform (0, 0.024) | 196 sequences from 8246 suspected P.1 cases as of 7 February 2021 |
| | Hong Kong | Uniform (0, 0.116) | 117 sequences from 1012 confirmed cases as of 7 May 2020 |

of 10 days[15]. A uniform distribution prior was used for the sampling proportion, which models changes in case ascertainment. Four independent MCMC chains were run for 50 million MCMC steps with sampling every 5000 steps for each dataset. These MCMC runs were combined using LogCombiner v2.6.5.[76] and the effective sample size of parameter estimates evaluated using TRACER v1.7.2[75]. We obtained an effective sample size above 200 for all parameters (indicating convergence) and plotted all results using the bdskytools R package (https://github.com/laduplessis/bdskytools).

**Estimation of growth rates.** For each dataset, a scaled proxy for $r_t$ was obtained from the *Skygrowth* method[77] within R. *Skygrowth* uses a non-parametric Bayesian approach to apply a first-order autoregressive stochastic process on the growth rate of the effective population size. The MCMC chains were run for one million iterations for each dataset on their MCC tree with an Exponential ($10^{-5}$) prior on the smoothing parameter. The *Skygrowth* model was parameterised assuming that the effective population size of SARS-COV-2 could change every two weeks. To facilitate a comparison of the scaled proxy for $r_t$ estimated by *Skygrowth* and exponential $r_t$ estimated by *EpiFilter*, the $r_t$ estimated by the *Skygrowth* method was rescaled to the exponential growth rate. This was achieved by adding a gamma rate variable to the scaled proxy for $r_t$, which assumed a mean duration of infection of 10 days[15], to calculate $R_t$. Subsequently, the Wallinga-Lipsitch equation (Eq. 2) was used to convert $R_t$ into the exponential growth rate[60].

**Comparing parameter estimates from genetic and epidemiological data**
To compare estimates derived from epidemiological and genetic data the Jensen-Shannon divergence ($D_{JS}$)[78], which measures the similarity between two probability mass functions (PMFs), was applied. The $D_{JS}$ offers a formal information theoretic evaluation of distributions and is more robust than comparing Bayesian credible intervals (BCIs) since it considers both the shape and spread of a given distribution. The $D_{JS}$ is a symmetric and smoothed version of the Kullback-Leibler divergence ($D_{KL}$) and is commonly used in the fields of machine learning and bioinformatics. The $D_{KL}$ between two PMFs, $P$ and $Q$, is defined in Eq. (3) below[79], with $x$ spanning the common domain of those PMFs.

$$D_{KL}(P||M) = \sum_x P(x) \log\left(\frac{P(x)}{Q(x)}\right) \quad (3)$$

To calculate the PMF for each epidemiological parameter, the cumulative probability density function was extracted for each model, converted to a probability density function and a discretisation procedure was applied to generate the associated PMF.

The Jensen-Shannon distance (JSD) is a metric which takes the square-root of the total $D_{JS}$ and is the metric that we used to compare

parameter estimations from differing sampling strategies. The JSD can be calculated using Eq. 4 with $P$ and $Q$ representing the two probability distributions and $D_{KL}$ as the KL divergence. A smaller JSD metric indicates that two probability distributions ($P$ and $Q$) are more similar with a Jensen-Shannon distance of 0 uniquely indicating that both distributions are equivalent. The mean JSD was taken over all intervals for the BDSKY and *Skygrowth* models to obtain an overall measure of the level of estimated similarity across the epidemic trajectory. We do not expect the JSD to perfectly align with the 95% highest posterior density interval if the shapes of distributions from different schemes are very different.

$$\text{JSD}(P||Q) = \sqrt{\frac{1}{2} D_{KL}(P||M) + \frac{1}{2} D_{KL}(Q||M)} \text{ where } M = \frac{1}{2}(P+Q) \quad (4)$$

**Reporting summary**
Further information on research design is available in the Nature Research Reporting Summary linked to this article.

## Data availability
All genomic data can be found here: https://www.gisaid.org/ (GISAID Acknowledgements in Supplementary Table 4). Hong Kong case data was taken from: https://www.chp.gov.hk/. Brazilian case counts were taken from the SIVEP-GRIPE database. Accession numbers of sequences used can be found within Supplementary Tables 2 and 3.

## Code availability
Code reproducing the analyses presented in this study are freely available at https://github.com/rhysinward/Phylodynamic-Subsampling.

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

## Acknowledgements

Role of the funding sources: N.R.F. acknowledges support from Wellcome Trust and Royal Society Sir Henry Dale Fellowship (204311/Z/16/Z), Bill and Melinda Gates Foundation (INV-034540) and Medical Research Council-Sao Paulo Research Foundation (FAPESP) CADDE partnership award (MR/S0195/1 and FAPESP 18/14389-0) (https://caddecentre.org). K.V.P. acknowledges support from grant reference MR/R015600/1, jointly funded by the UK Medical Research Council (MRC) and the UK Department for International Development (DFID) and from the NIHR Health Protection Research Unit in Behavioural Science and Evaluation at University of Bristol. N.R.F. and K.V.P. acknowledge funding from the MRC Centre for Global Infectious Disease Analysis (reference MR/R015600/1), which is jointly funded by the UK Medical Research Council (MRC) and the UK Foreign, Commonwealth & Development Office (FCDO), under the MRC/FCDO Concordat agreement and is also part of the EDCTP2 programme supported by the European Union; and acknowledge funding by Community Jameel. R.P.D.I acknowledges support from European Union Horizon 2020 project MOOD (#874850).

## Author contributions

R.P.D.I., K.V.P., and N.R.F. conceived and designed the study, R.P.D.I. wrote and performed the analyses. R.P.D.I. wrote the manuscript which was edited and supervised by K.V.P. and N.R.F. All authors have contributed to and approved the manuscript for submission.

## Competing interests

The authors declare no competing interests.
