## [Peer Review File · Nature Communications]

Using multiple sampling strategies to estimate SARS-CoV-2 epidemiological parameters from genomic sequencing dataREVIEWER COMMENTS

Reviewer #1 (Remarks to the Author):

Line 234: Equation 1 has sigma while Table 1 has delta (Line 241).

Line 345: Should be "An HKY..."

Lines 406-414: Paragraph and Equation 4 are hard to parse with potential typos. Was the first "(PDF)" supposed to be "(CDF)" or did you intend to abbreviate twice. Using tau for both the PDF and CDF (if I am reading that right), is quite confusing. In the equation, the relationship between t , s , ν , and i are very unclear.

Line 408: Is there supposed to be a sentence break on line 408?

Line 495: Using "s/s/y" before defining the abbreviation on line 498.

Line 534 & Figure 3: The Figure 1C violin plot doesn't appear to support the R_0 of 3.67 mentioned on line 534. The figure and caption don't mention what the violin plots or the marked lines mean. Is the thick line posterior mean and the thinner lines the credible interval?

Figures 3-7, Lines 312-318: Make the sampling scheme ordering consistent. I would have the E panels in Figures 4-7 have the same sampling scheme order as their A-D panels which matches Figure 3. Lines 312-318 also have a different ordering than the rest.

Line 978: If there's room, I'd want Supplementary Table 1 in the Introduction.

Reviewer #2 (Remarks to the Author):

Rhys et al. address an interesting and important question relating to the best sampling strategies for genomic sequences of SARS-CoV-2. Using sequences to estimate epidemiological parameters from genomic data requires determining how sequence sampling can influence these parameters. This is particularly important for resource-poor settings where sequencing capabilities are limited. They find that the sampling strategies can induce biases particularly in the estimates of R_t and r_t that are sensitive to changes in sampling whilst R_0 and the date of origin of a lineage are relatively robust to different sampling strategies.

The authors use the Jensen-Shannon Distance to compare the parameters obtained from the different genomic sampling strategies to those obtained using epidemiological data, however it is difficult to see how the different genomic sampling strategies compare to each other. For example, Figure 5 shows that the JD distance of the proportional sampling and uniform sampling to Epifilter are the same but this does not necessarily mean that the proportional sampling and uniform sampling are the same. Do sampling strategies show the same wrong pattern? I would suggest presenting the results as a matrix of pairwise JD distances so this information can be conveyed.

The manuscript also contains several misspellings, confusing figure captions and missing methodological information:

Line 270 – what is defined as high quality complete genomes?

Line 280: selected for. > selected.

Line 284: which version of Pangolin software was used?

Line 285 selected for. => selected.

Line 323: as was used => and was used

Line 492:

This is likely due to the Hong Kong datasets have a wider sampling interval => having

Line 505: overlapped -> overlapping

Seeing the confirmed cases from Figure 2 alongside the Rt estimated in Figure 4 would make comparison easier.

Figure 5 is unclear "The solid line represents the mean rt estimate with Skygrowth in red and BDSKY in blue. " => Isn't the skygrowth shown in blue?

Figure 7 caption seem inconsistent with the legend.

In Fig4 it seems quite clear that Proportional is optimal – are the other three sampling strategies more similar to each other than they are to the proportional. In Figures 5-7, is the distance between each sampling strategy very distinct to the unsampled?

Line 640: than initial => than the initial

Line 675: remove "through"

Supplementary figure 1: Has this been limited to a specific geographical region? Amazonas only?
Add the details to the caption.

Point-by-point Response to Reviewers – Manuscript ‘Using multiple sampling strategies to estimate SARS-CoV-2 epidemiological parameters from genomic sequencing data’

We thank both the reviewers and editor for their constructive comments. Here we summarise the changes that we have made for our resubmission, followed by point-by-point responses to the reviewers. We also include a version of the revised manuscript with tracked changes.

Typing errors have been corrected and clarification regarding the Jensen Shannon divergence and violin plots have been given. A matrix highlighting the similarities among estimates derived under different sampling strategies has been included in panel F within Figures 4-7 and appropriate discussion of its consequences to our analysis are included. We highlight here in blue the text what has been added to the manuscript.

Reviewer #1 (Remarks to the Author):

Query 1. Line 234: Equation 1 has sigma while Table 1 has delta (Line 241).

Reply: We have now amended this equation.

Query 2. Line 345: Should be "An HKY..."

Reply: Thanks, we have made this correction.

Query 3. Lines 406-414: Paragraph and Equation 4 are hard to parse with potential typos. Was the first "(PDF)" supposed to be "(CDF)" or did you intend to abbreviate twice. Using tau for both the PDF and CDF (if I am reading that right), is quite confusing. In the equation, the relationship between t, s, nu, and i are very unclear.

Reply: To improve clarity to the reader we have removed the equation and instead explain the procedure with descriptive text. This now reads: “To calculate the PMF for each epidemiological parameter, the cumulative probability density function was extracted for each model, converted to a probability density function and a discretisation procedure was applied to generate the associated PMF.”

Query 4. Line 408: Is there supposed to be a sentence break on line 408?

Reply: We have removed this break.

Query 5. Line 495: Using "s/s/y" before defining the abbreviation on line 498.

Reply: This has been corrected within text.

Query 6. Line 534 & Figure 3: The Figure 1C violin plot doesn't appear to support the R0 of 3.67 mentioned on line 534. The figure and caption don't mention what the violin plots or the marked lines mean. Is the thick line posterior mean and the thinner lines the credible interval?

Reply: We have amended the text and figure caption to improve clarity. The text now reads ‘We found from using genomic data, Hong Kong had a posterior mean R_0 estimate of 2.07 (Figure 3A) across all sampling strategies. Using a proportional sampling strategy gave the highest posterior mean R_0 estimate of 2.38 with the unsampled sampling strategy giving the lowest posterior mean R_0 estimate of 1.87. Overall, Brazil had a higher posterior mean R_0 estimate with a value of 2.24 (Figure 3B) across all sampling strategies. The uniform sampling strategy yielded the highest posterior mean R_0 estimate of 2.50 while the unsampled sampling strategy gave the lowest one of 1.82. Using case data, we found similarly found that Hong Kong had a lower R_0 of 2.17 (95% credible interval (CI) = 1.43 - 2.83) when compared to Amazonas which had a R_0 of 3.67 (95% CI = 2.83 – 4.48). All sampling schemes for both datasets were characterised by similar R_0 values (Figure 3) indicating that the estimation of R_0 is robust to changes in sampling scheme.’ Moreover, with the figure caption of each violin plot we have stated that ‘The central line represents the posterior mean estimate and intervals demarcate the 95% Highest Posterior Density Interval.’ to improve clarification.

Query 7. Figures 3-7, Lines 312-318: Make the sampling scheme ordering consistent. I would have the E panels in Figures 4-7 have the same sampling scheme order as their A-D panels which matches Figure 3. Lines 312-318 also have a different ordering than the rest.

Reply: We thank the reviewer for their comment and the ordering of the sampling schemes have been changed to reflect our figures. With respect to the figure E panels, we would prefer to keep the existing order to reflect the ranking from lowest to highest JSD as we feel this improves interpretation.

Query 8. Line 978: If there's room, I'd want Supplementary Table 1 in the Introduction.

Reply: Thanks, we have placed the supplementary table in the introduction.

Reviewer #2 (Remarks to the Author):

Rhys et al. address an interesting and important question relating to the best sampling strategies for genomic sequences of SARS-CoV-2. Using sequences to estimate epidemiological parameters from genomic data requires determining how sequence sampling can influence these parameters. This is particularly important for resource-poor settings where sequencing capabilities are limited. They find that the sampling strategies can induce biases particularly in the estimates of R_t and r_t that are sensitive to changes in sampling whilst R_0 and the date of origin of a lineage are relatively robust to different sampling strategies.

We thank the Reviewer for the positive assessment of our work.

Query 1. The authors use the Jensen-Shannon Distance to compare the parameters obtained from the different genomic sampling strategies to those obtained using epidemiological data, however it is difficult to see how the different genomic sampling strategies compare to each other. For example, Figure 5 shows that the JD distance of the proportional sampling and uniform sampling to Epifilter are the same but this does not necessarily mean that the proportional sampling and uniform sampling are the same. Do sampling strategies show the

same wrong pattern? I would suggest presenting the results as a matrix of pairwise JD distances so this information can be conveyed.

Reply: Thanks for this excellent suggestion, we hope that these additional analyses have improved the understanding and clarity of our study. To determine if the sampling strategies were showing the same wrong pattern, as suggested, we computed a matrix of pairwise JSD. These are now included in Figures 4-7 (panel F) for each pair of sampling strategies: e.g., unsampled vs proportional, proportion vs uniform, etc. We found that the unsampled sampling scheme was consistently distinct from all other sampling schemes, whilst the uniform and inverse sampling schemes were consistently the most similar. Also see our response to Query 12 below.

Query 2. Line 270 – what is defined as high quality complete genomes?

Reply: We consider high-quality genomes to contain <1% N, or non-identified nucleotides and to be complete they must have a genome bigger than >29 kb. We have included this point in the main text now.

Query 3. Line 280: selected for. > selected.

Reply: Thanks, correction made.

Query 4. Line 284: which version of Pangolin software was used?

Reply: We have now included the specific Pangolin version.

Query 5. Line 285 selected for. => selected.

Reply: Amendment made.

Query 6. Line 323: as was used => and was used

Reply: Thanks, fixed now.

Query 7. Line 492: This is likely due to the Hong Kong datasets have a wider sampling interval => having

Reply: We have now amended this sentence.

Query 8. Line 505: overlapped -> overlapping

Reply: Thanks, also fixed now.

Query 9. Seeing the confirmed cases from Figure 2 alongside the Rt estimated in Figure 4 would make comparison easier.

Reply: Thanks, for the suggestion. Cases have now been added to Figures 4-7 to enable easier comparison between epidemiological parameters and case counts.

Query 10. Figure 5 is unclear “The solid line represents the mean r_t estimate with Skygrowth in red and BDSKY in blue. “ => Isn't the skygrowth shown in blue?

Reply: Thanks for pointing out this error, it is now corrected in the figure caption.

Query 11. Figure 7 caption seems inconsistent with the legend.

Reply: We have updated the caption to read: “The solid line represents the mean \$r_t\$ estimate with *EpiFilter* in orange and *Skygrowth* in blue”.

Query 12. In Fig4 it seems quite clear that Proportional is optimal – are the other three sampling strategies more similar to each other than they are to the proportional. In Figures 5-7, is the distance between each sampling strategy very distinct to the unsampled?

Reply: Thanks for the suggestion. To determine if the JSD between each sampling strategy is distinct compared to the unsampled case we have included a matrix of pairwise JSD values in panel F of Figures 4-7 for every pair of sampling strategies (also see our response to Query 1 above). We discuss the results of this in the main text as: ‘We found that estimates from all sampling schemes were distinct from those obtained using the unsampled data and that on some instances the sampling schemes were also appreciably different from one another (see panel F in Figures 4-7) with the uniform and reciprocal-proportional sampling strategies being most similar. This highlights how different sampling schemes can produce significantly differing estimates of epidemiological parameters and underscores the need for considering sampling and its potential impact on estimations.’.

Query 13. Line 640: than initial => than the initial

Reply: We have now amended this sentence.

Query 14. Line 675: remove “through”

Reply: Thanks, removed now.

Query 15. Supplementary figure 1: Has this been limited to a specific geographical region? Amazonas only? Add the details to the caption.

Reply: We have now amended this figure caption stating that it has been limited to only the Amazonas state of Brazil.